# Technical and Tactical Performance in Women’s Singles Pickleball: A Notational Analysis of Key Match Indicators

**DOI:** 10.3390/jfmk10010020

**Published:** 2025-01-03

**Authors:** Iván Prieto-Lage, Xoana Reguera-López-de-la-Osa, Christopher Vázquez-Estévez, Alfonso Gutiérrez-Santiago

**Affiliations:** 1Observational Research Group, Faculty of Education and Sport, University of Vigo, 36005 Pontevedra, Spain; ivanprieto@uvigo.es (I.P.-L.); christopher.vazquez@uvigo.gal (C.V.-E.); 2Education, Physical Activity and Health Research Group (Gies10-DE3), Galicia Sur Health Institute (IIS Galicia Sur), SERGAS-UVIGO, 36208 Vigo, Spain

**Keywords:** performance, match analysis, racket sports, forehand, backhand

## Abstract

**Background:** Pickleball has experienced remarkable growth in recent years, yet studies exploring its specific characteristics are scarce. This investigation provides a detailed notational analysis of women’s singles pickleball, evaluating the technical and tactical performance indicators in the game. **Method:** An observational methodology was used to analyze all points from five PPA Tour tournaments. The matches were recorded and coded using LINCE PLUS software, version 2.1.0, with a category system designed for this sport. A descriptive analysis was conducted with IBM SPSS version 25.0, and Theme 6.0 Edu software was used to detect gameplay patterns. The statistical significance was set at *p* < 0.05. **Results:** The findings indicate that serving players have a slight advantage, winning 55.1% of points. Most of the points were resolved through unforced errors, accounting for 63.7% of the total, primarily from forehand strokes in short rallies and backhand strokes in medium rallies. The most frequent hitting zones for point termination were near the non-volley zone (35.8%) and behind the baseline (38.6%). **Conclusions:** This study provides a deeper understanding of performance in women’s pickleball, highlighting technical and tactical patterns that offer guidelines for optimizing strategies and techniques in the sport.

## 1. Introduction

Pickleball, although a relatively young sport created in 1965, has rapidly gained popularity and is now considered one of the fastest-growing sports in the United States and globally. This growth is largely due to its unique combination of elements from tennis, badminton, and table tennis [1]. Among racquet sports, pickleball has quickly established itself as a prominent recreational activity, surpassing other traditional racquet sports in terms of growth rates, particularly in community centers and recreational leagues [2]. It is especially popular among older adults, making up a significant portion of its player base. This demographic is drawn to the sport due to its low-impact nature, making it accessible and ideal for people aged 50 and above, a characteristic that differentiates it from other sports like tennis [3]. As the sport continues to grow in popularity [4], it is essential to deepen the understanding of the dynamics that define on-court performance, as has been done with similar sports [5,6,7]. However, despite its expansion, pickleball has received little attention from the perspective of notational analysis, especially in identifying technical–tactical patterns that explain its strategic nature and the style of play. One of the most interesting aspects of these analyses is the study of forehand and backhand execution and how these strokes can influence the effectiveness of the game [8,9].

In racket sports, the ability to execute shots from both sides of the body is a determining factor in competitive success. Traditionally, it has been observed that players tend to be more effective and comfortable with their forehand, which often results in a higher number of winners from that side [10,11]. However, this preference also exposes a vulnerability: reliance on a single stroke can be exploited by opponents who consistently target the player’s backhand, a shot that is commonly perceived as less powerful or less accurate [12].

In the context of pickleball, although there is limited prior research, the analysis of strokes takes on special importance due to the unique characteristics of the game, such as the smaller court size and the constant proximity of players to the net [13]. Both shots from the back of the court and those made in the non-volley zone are crucial in determining the outcome of each point. The ability to execute both forehand and backhand strokes with precision, as well as volleys on both sides, becomes a key performance factor, similar to what has been observed in sports like tennis and padel [14,15].

The analysis of gameplay patterns in women’s competitions in other racket and paddle sports has revealed a tendency to make more unforced errors with the backhand, especially in pressure situations, where the player is forced to hit from an uncomfortable position [16,17,18]. This asymmetry in stroke execution can influence not only the number of unforced errors but also the effectiveness of attacking shots, i.e., winners [19]. It is expected that winners are generated more frequently from the forehand side, highlighting the need to develop a training strategy that addresses this disparity. A possible explanation could be related to the similarity between the kinematics of the humerus and forearm in the throwing motion during childhood [20], which is more similar to the mechanics of the forehand stroke than the backhand.

In addition to the individual execution of strokes, the direction in which the ball is sent plays a crucial role. In racket sports, it is common for players to direct their shots towards the opponent’s backhand, aiming to exploit the relative weakness on that side [21]. This tactical pattern not only reinforces the importance of effective symmetry in play but also highlights the need for robust defensive strategies to counter such tactics.

In this context, symmetry in pickleball not only refers to the balance in executing forehand and backhand shots but also to the ability to maintain equilibrium in producing winners and minimizing unforced errors from both sides of the court. The current research aims to explore how these dynamics of symmetry or asymmetry influence player performance, analyzing factors such as rally duration, the court areas where shots are executed, and the strategies employed to exploit the opponent’s weaknesses. Understanding these patterns is crucial for advancing knowledge in pickleball, a sport experiencing rapid growth but with limited scientific research. The findings have the potential to provide valuable insights for coaches and players, guiding the development of evidence-based training programs to enhance performance and reduce technical deficiencies.

This study hypothesizes that (1) asymmetries in stroke execution, particularly in the backhand, will be associated with a higher occurrence of unforced errors; (2) specific court zones, such as the non-volley zone and baseline, will play a pivotal role in point resolution; and (3) serving players will exhibit a slight advantage in winning points, as observed in other racket sports. By addressing these aspects, this research seeks to fill a critical gap in the literature, contributing to the growing understanding of technical and tactical factors in elite women’s pickleball.

This approach will not only provide a better understanding of the technical and tactical skills required for success in women’s pickleball but also identify key areas for improvement that could be addressed through specific training programs. Ultimately, this study seeks to offer a novel perspective on the importance of symmetry in sports performance, providing coaches and players with concrete tools to optimize their performance on the court.

## 2. Materials and Methods

### 2.1. Sample

The analysis in this research focused on the points played during the examined pickleball matches. Specifically, data were collected from five tournaments (Las Vegas, Cincinnati, Kansas, Seattle, and Denver), resulting in the evaluation of 15 matches and a total of 1145 points. A total of 17 different players were observed across the analyzed tournaments, as some players reached the semifinals multiple times. The participants included professional players who had advanced to at least the semifinal stage in these events, representing a high level of expertise in the sport. These elite players were selected based on their performance in the professional circuit, ensuring that the analysis captured high-quality and competitive gameplay. As the study relied on publicly available match footage and did not involve experimental procedures, obtaining informed consent was deemed unnecessary [22]. Ethical approval was granted by the Ethics Committee of the Faculty of Education and Sport Sciences at the University of Vigo (application 07-280722).

### 2.2. Design

This study examines the patterns of play in women’s singles pickleball through an observational approach [23]. The research design employed [24] follows a nomothetic framework, encompassing every point played in the semifinals and finals of five PPA Pro Tour tournaments during the 2023 season. Furthermore, it adopts a longitudinal perspective, spanning the entire season, and is unidimensional, as it does not account for concurrent behaviors during analysis.

### 2.3. Observational Instruments and Technological Tools

The observational tool used in this study was the OI-PICKLEBALL-S23 (see Table 1 and Figure 1), a category system specifically designed for analyzing pickleball during the 2023 season. This instrument was previously applied in a study focused on the men’s category [25], where it underwent a validation process to ensure its reliability and alignment with the research objectives. The validation included alignment with the theoretical framework and consultation with three experts in racket and/or paddle sports and observational methodology, who showed over 95% agreement with the instrument. The data were recorded using LINCE PLUS software, version 2.1.0 [26].

To ensure the highest level of accuracy in the preparation of this manuscript, the authors used ChatGPT (OpenAI, San Francisco, CA, USA) for assistance with translating and refining the technical terminology. This tool was employed solely to enhance linguistic clarity and did not contribute to the scientific content or the interpretation of the data. Additionally, the Consensus platform was utilized to efficiently identify and select relevant scientific literature supporting the discussion of our findings.

### 2.4. Procedure

The data collection process involved locating, downloading, and reviewing the semifinals and finals from the five selected tournaments. Prior to performing the data quality assessments, two experts in pickleball and observational methodology underwent specialized training in the use of the observational instrument. This training, conducted over nine two-hour sessions across three weeks, included familiarization with both the instrument and the LINCE PLUS recording software (version 2.1.0), using videos of men’s pickleball matches from the 2022 season.

To ensure the precision of the data recording [27], the data quality was evaluated by calculating intra-observer and inter-observer reliability using the Kappa coefficient [28] with the LINCE PLUS software, version 2.1.0. These evaluations were based on a subset of points not included in the final sample (*n* = 200; 10% of the total sample). The intra-observer Kappa values were 0.96 for the first observer and 0.99 for the second, while the inter-observer Kappa value was 0.98. Following the validation of the data quality, the second observer analyzed all the points in the study sample. Afterward, the recorded data were compiled into an Excel file, providing a detailed sequence of actions for each analyzed point. This file was designed to facilitate the seamless transfer of the data into formats compatible with IBM-SPSS version 25 and THEME version 6 Edu, the software used for the study’s statistical analyses.

### 2.5. Data Analysis

The descriptive analysis for this study was performed using the Statistical Package for the Social Sciences (SPSS), version 25.0 (IBM-SPSS Inc., Chicago, IL, USA), with the statistical significance set at *p* < 0.05.

To examine differences within the categories of each criterion from the observational instrument, a *χ*^2^ goodness-of-fit test was conducted. Additionally, a *χ*^2^ test of independence was applied through cross-tabulation to analyze the relationship between the point ending and final stroke criteria.

Game patterns were identified through a T-Pattern analysis using THEME 6 Edu software (PatternVision Ltd., Reykjavik, Iceland) [29]. This method detects recurring behavioral patterns within a temporal sequence based on statistical probabilities [30]. The software also enables the identification of sequential structures using the order parameter. The analysis adhered to the following criteria: (a) the presence of at least three T-Patterns within the observed sequence; (b) a 90% redundancy reduction to minimize the repetition of similar T-Patterns; and (c) a significance threshold of 0.005.

## 3. Results

### Descriptive Analysis

Table 2 provides a descriptive analysis of the study, including the *χ*^2^ goodness-of-fit test conducted for the categories within each criterion.

The study revealed that service faults were minimal in this sport, representing only 2.3% of cases. Most points were resolved in rallies of short (42.1%) or medium duration (44.5%). The zones where the final stroke was executed were mainly concentrated near the non-volley zone, specifically zone 2 (35.8%) and zone 4, located behind the baseline (38.6%). Additionally, a high percentage of points ended in unforced errors (63.7%). Regarding winners, they were mostly directed towards the middle and the back of the court, with a tendency to target the opponent’s theoretical forehand zone, particularly in zones 4 (8.8%) and 6 (10.6%).

The server won more points than the returner, and both players gained most of their points through unforced errors by their opponent. Regarding the final shot that determined the points (whether winners or unforced errors), forehand strokes (36.1%) and backhand strokes (30.5%) stood out, as well as forehand and backhand volleys (15.3% and 12.7%, respectively). No other type of shot exceeded 3% in occurrence.

Figure 2 presents an analysis of the relationship between the final stroke and the point ending based on the type of rally, conducted through a *χ*^2^ test of independence.

The data revealed significant differences between the three types of rallies analyzed. In short rallies, most points were decided through unforced errors, which were more frequent from the returner, especially with forehand strokes, although numerous backhand errors were also recorded. Points won by the returner were mainly due to unforced errors from the opponent’s forehand, although the backhand also contributed significantly to errors. Regarding points won on a serve, both forehand and backhand strokes were used, while winning volleys were almost non-existent. In contrast, among the returners, forehand volleys were predominant.

In medium-length rallies, the resolution pattern was different. Although unforced errors remained the primary cause of points, the percentages between the server and the returner were similar. In both cases, errors were predominantly preceded by backhand strokes. Additionally, a significant number of points concluded with winners, with forehand volleys being prominent for both players, followed by the server’s forehand strokes and the returner’s backhand volleys.

In long rallies, the server mainly won points through winners, with an almost equal distribution between forehand volleys and forehand groundstrokes. On the other hand, the points won by the returner were mostly due to unforced errors, with forehand mistakes being predominant, followed closely by backhand errors. Regarding winners from the returner, forehand volleys stood out.

Table 3 and Table 4 present a T-Pattern-based analysis of the probability of winning the point based on the rally type and the final stroke, also considering the hitting zone and the player who won the point. In Table 3, points where a service fault occurred were excluded (*n* = 22). Both in Table 3 and Table 4, points that ended with strokes other than forehand, backhand, forehand volley, or backhand volley were omitted due to their residual nature and in order to simplify the analysis (*n* = 7 and *n* = 15, respectively).

The T-Patterns obtained for the analysis of short rallies revealed that most points were resolved through unforced errors from zone 4, following both forehand and backhand shots, accounting for 58.2% of the total points. For the server, points gained from opponent errors were distributed as 25.8% for forehand errors and 16.1% for backhand errors. For the receiver, opponent errors contributed 8.9% of the points from forehand errors and 7.4% from backhand errors. Additionally, winners by the receiver were observed, with backhand volleys executed from zone 2 standing out, accounting for 3.2% of the points.

For the points extending between 5 and 9 shots, the server achieved a significant number of points from zone 2, primarily with forehand volleys (7.1%) and backhand volleys (4.2%). By keeping the opponent in zone 4, the server successfully induced several backhand errors (5.2%), though this strategy was less effective against the opponent’s forehand (3.3%). For the returner, the points gained from forced errors by the opponent were noteworthy, with errors from forehand (4.7%) and backhand (4.9%), while keeping the opponent in zone 4. A similar pattern was observed in zone 3, though to a lesser extent (2.4% and 3.8%, respectively). Additionally, from zone 2, the returner achieved a significant number of points with winners, using both forehand volleys (5.9%) and backhand volleys (4.2%), and also forced errors from the opponent with these same strokes (5.2% and 4.2%, respectively).

For points lasting between 5 and 8 shots, the server achieved a notable number of points from zone 2, primarily through forehand volleys (7.1%) and backhand volleys (4.2%). It was observed that by keeping the opponent in zone 4, the server caused their rival to commit numerous backhand errors (5.2%), although this strategy was less effective against the opponent’s forehand (3.3%). For the receiver, when managing to keep their opponent in zone 4, the points won from unforced errors by the opponent were particularly prominent, both from forehand errors (4.7%) and backhand errors (4.9%). A similar pattern was identified in zone 3, though to a lesser extent (2.4% and 3.8%, respectively). Additionally, from zone 2, the receiver achieved a significant number of points through winners, using both forehand volleys (5.9%) and backhand volleys (4.2%), and also generating errors from their opponent with these same shots (5.2% and 4.2%, respectively).

## 4. Discussion

The purpose of this research was to carry out a detailed notational analysis of women’s pickleball at its highest level, focusing on the key technical and tactical aspects of the game. This study has provided a comprehensive view of the patterns of play, highlighting the importance of hitting zones and the types of strokes that determine the outcome of points depending on the type of rally.

This study revealed that, unlike other racket sports such as tennis, where the serve can be a decisive factor [31,32], pickleball shows a relatively high percentage of first serves in play (97.7%) but no direct service points, similar to findings in men’s pickleball [25]. This is partly due to the sport’s rules, which require an underhand serve, resulting in a slower start to the point. The slight advantage gained by the server in women’s games (55.1%) differs from the men’s data, where serving players won only 46.6% of points, indicating a less pronounced server advantage in men’s pickleball. This variation in serve effectiveness may reflect gender-related differences in the serving technique or overall court dynamics, warranting further exploration in future studies. The serve appears to have a greater impact on match outcomes in women’s pickleball compared to men’s, although in both cases, it remains less decisive than in sports like tennis or padel [6,33,34,35]. These findings highlight the potential benefits of optimizing the serve to improve performance in both genders.

The results from this study suggest that, in women’s pickleball, points tend to be decided near the net (zone 2) and from behind the baseline (zone 4). This reflects a combination of baseline strategies more typical of tennis [21] and net-approach tactics characteristic of padel [36,37]. In women’s matches, 35.8% of the finishing shots occurred in zone 2 (near the non-volley zone) and 38.6% in zone 4 (behind the baseline). In contrast, men’s matches exhibited a stronger reliance on net play, with 50.7% of finishing shots in zone 2, while only 26.6% occurred in zone 4 [25]. This comparison suggests that women tend to rely more on baseline strategies, while men prioritize net approaches to a greater extent. Despite these differences, both genders demonstrate the importance of maintaining a balanced strategy that leverages the tactical advantages of both baseline and net play. The T-Pattern analysis from both studies underscores the significance of approaching the net to finish points, reinforcing the hybrid tactical nature of pickleball.

When looking at the final stroke, differences between the genders were observed. In women, the percentage of backhand volleys as the final stroke was 12.7%, while in men it was 17.7%. Women finished 30.5% of points with a backhand, compared to 21.7% in men. The forehand was used to finish 36.1% of points in women and 34.4% in men. Finally, forehand volleys represented 15.3% in women and 20.7% in men [25]. This suggests that while both genders rely heavily on forehands and backhands to conclude points, men are more aggressive in using volleys, particularly forehand volleys, for point finishes.

These differences in the final strokes suggest that women tend to rely more on backhand strokes to close out points, while men prefer forehand and backhand volleys. This could have strategic implications in the game, as the higher use of forehand volleys in men (20.7%) might reflect greater aggressiveness and a focus on net play, potentially leading to faster points and a more offensive strategy. In contrast, the greater reliance of women on backhand strokes (30.5%) could indicate a more defensive approach or a need for longer rallies before finishing the point. These differences in the final strokes may influence training strategies, suggesting that women could benefit from working on optimizing their forehand volleys to increase the speed and effectiveness of finishing points.

The high incidence of unforced errors (63.7%) observed in this study is comparable to the percentages recorded in other racket sports such as tennis and padel [6,15,21,38], underscoring the critical role of precision in gameplay. In women’s matches, backhand errors were more frequent during medium rallies, while forehand errors dominated shorter rallies. This finding is consistent with studies of other racket sports [16,17,36,39], where unforced errors tend to increase with the backhand, suggesting the need for improvement in backhand stroke technique. Similarly, in men’s pickleball [25], unforced errors also played a significant role, accounting for 58.3% of points. These observations emphasize the importance of targeted training to enhance stroke consistency and accuracy, particularly focusing on backhand technique, to minimize errors and optimize performance across both genders.

Winners’ shots in women’s pickleball were mainly concentrated in forehands and backhands, with a notable presence of volleys by the returner in short rallies or by both players in longer rallies. The low incidence of other types of finishing shots (less than 3%) indicates that forehands, backhands, and volleys are the most effective to close out points. This pattern shows both similarities and differences with other racket sports. For example, in padel, volleys are a key tool to finish points, but the smash also plays an important role [15,39], something that does not occur as frequently in pickleball. On the other hand, in singles tennis, groundstrokes are the predominant strokes to end points, while volleys are less common [21,40].

Women generated 36.3% of their points through winners, while men generated 41.6% [25]. This indicates that men are slightly more efficient at producing winners than women. This difference suggests that women may benefit from developing a more offensive mindset or optimizing their offensive shots, particularly in situations where men tend to generate winners with volleys and more aggressive groundstrokes. This difference suggests that women may need to work on improving their ability to finish points more consistently, potentially by increasing their offensive play or optimizing their shot selection. The lower percentage of winners in women’s pickleball could be related to tactical decisions, such as a stronger reliance on rallies and consistency, or it could reflect the need to refine specific strokes that can help generate more winners, such as forehand volleys or groundstrokes.

In tennis, winners are often achieved through groundstrokes from the back of the court, where the power and precision of forehands and backhands are fundamental to securing points [21,41]. In contrast, padel is characterized by a predominance of volleys, due to the importance of taking the net to control play and look for opponent errors [37,42].

In women’s pickleball, just like in padel, when a player successfully positions themselves to execute a volley, the likelihood of winning the point increases significantly, especially in short rallies while returning or in medium rallies while serving. However, it also shows a significant reliance on forehands and backhands in longer rally situations, similar to tennis. This suggests that women’s pickleball combines elements of both sports, highlighting the importance of perfecting both groundstrokes and volleys. When comparing this with men’s pickleball, it becomes clear that while both genders rely on groundstrokes for finishing points, men tend to exhibit a greater dependence on volleys as final strokes (20.7%), compared to women (15.3%). This greater emphasis on volleys in men’s play aligns more closely with padel, where net play and volleying are more prominent. In contrast, the women’s game leans slightly more toward tennis-like strategies, with a stronger focus on groundstrokes for the point conclusion.

### 4.1. Practical Implications

Short possessions training: Working with shorter possessions can help players improve their decision making under pressure and execute fast offenses, such as counterattacks. This type of training maximizes the likelihood of success in high-pressure moments. Considering gender differences, it may be particularly beneficial for women to focus on improving forehand volleys and fast transitions in short rallies, as these are less utilized compared to men’s game. In contrast, men’s training should further enhance both forehand and backhand volleys, given their stronger reliance on net play.

Strategy based on the score: Coaches can design drills that simulate different scoring situations. For example, players who are behind in critical moments should focus on faster attacks with fewer shots, while players who are ahead could benefit from longer possessions to control the pace. In women’s pickleball, focusing on creating more offensive opportunities through forehand volleys could help increase point completions, especially in fast-paced situations. For men, who rely more on net play, drills can focus on controlling the net and executing effective volleys in these critical moments.

Consistency and unforced errors: The high incidence of unforced errors (63.7%) underscores the need for players to focus on reducing errors, particularly with the backhand and forehand during short and medium rallies. In women’s pickleball, improving backhand consistency could be a focal point, as unforced errors from this stroke were more frequent in medium rallies. For men, working on both forehand and backhand volleys can help reduce errors, particularly in the faster rallies’ characteristic net play.

Volley technique: Volleys, both forehand and backhand, represented a significant percentage of the decisive points. Training should, therefore, prioritize volley technique, particularly in the context of fast exchanges at the net. For women, given the slightly lower reliance on volleys compared to men, it would be beneficial to emphasize developing this skill in training, especially to increase offensive play in short and medium rallies. In men’s play, where volleying is a more dominant strategy, improving consistency and precision with volleys should be a key area of focus.

Movement optimization near the net and baseline: Since most points are decided near the net (zone 2) and from behind the baseline (zone 4), players need to optimize their movements in these areas to improve both defense and offense. Women may benefit from enhancing baseline play and positioning, especially given their reliance on longer rallies and more baseline-focused strategies. Men, however, may focus on improving their net coverage and quick transitions to better execute volleys and finish points.

Long rallies and winner generation: This study shows that points in long rallies are often resolved with winners or unforced errors, especially from the forehand. Players should focus on reducing errors during these rallies and learn to effectively transition to offense. Given the gender differences in shot selection during long rallies, training for women should focus on consistency and shot placement, particularly in generating winners from groundstrokes. For men, training should focus on finding opportunities for faster finishes, leveraging volleys and offensive groundstrokes.

Serve optimization: Although service faults were minimal, the server won more points than the receiver, which suggests the service remains a key element in determining match outcomes. Therefore, improving the serve should remain a priority. For women, given the slightly higher server advantage (55.1%) compared to men (46.6%), enhancing serving techniques and consistency could have a more pronounced impact on match outcomes.

### 4.2. Limitations and Future Perspectives

This study collected data from the semifinals and finals of five professional pickleball tournaments, excluding earlier rounds, which might have yielded different results. Since the data come exclusively from professional players, the findings may not be generalizable to other levels of play.

Furthermore, weather factors, which can influence player performance, were not considered, preventing a standardized representation across all pickleball matches. The primary focus on asymmetry within the game analysis may not comprehensively capture other critical factors for understanding the game.

While patterns of stroke and court positioning have been studied, the analysis may have fallen short in evaluating tactical decision making and the adaptations players make during match play. Additionally, numerical data on point duration or court zones may not fully reflect the underlying strategic dynamics.

Future research could delve into the relationship between physical conditioning and the ability to minimize unforced errors in long rallies. It would also be valuable to explore how biomechanics impact both backhand and forehand strokes, optimizing technique in both aspects.

Additional future perspectives include a deeper analysis of match temporality, assessing effort and rest times. This information could be crucial for designing more effective and specific training programs for players, enhancing their performance throughout matches.

## 5. Conclusions

This study provides a comprehensive view of elite women’s pickleball performance in the singles category, revealing gameplay patterns that enhance the understanding of the sport’s specific tactical and technical characteristics. The data analysis showed that most points were resolved in short and medium rallies, with unforced errors playing a significant role in determining points—particularly those made with forehand and backhand shots in short rallies and with backhand shots in medium rallies. This suggests specific areas of vulnerability in stroke technique that could be targeted for improvement.

The servers exhibited a slight advantage by scoring more points compared to the receivers, highlighting the importance of an effective serve within the overall game strategy. However, the high frequency of unforced errors, which reached 63.7%, indicates substantial opportunities for improving consistency and accuracy in shots.

The analysis also revealed that the predominant final stroke zones were near the non-volley zone and behind the baseline. This suggests that players tend to keep their opponents in disadvantageous positions to execute decisive shots. Additionally, the final strokes in women’s pickleball are primarily concentrated in forehand, backhand, forehand volley, and backhand volley shots, with other types of final strokes being virtually non-existent. This trend highlights the need to focus on optimizing these specific shots to enhance performance in professional-level matches.

## Figures and Tables

**Figure 1 jfmk-10-00020-f001:**
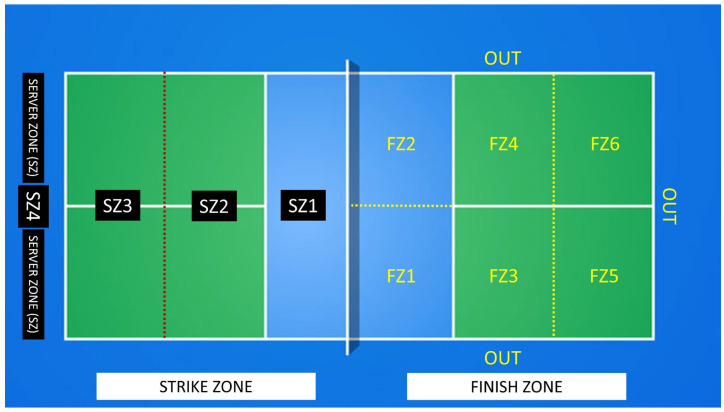
Visual representation of striking and finish zones on the court. The white lines represent the official pickleball court lines (with the net positioned in the center). The dashed yellow lines and the dashed red line are imaginary lines.

**Figure 2 jfmk-10-00020-f002:**
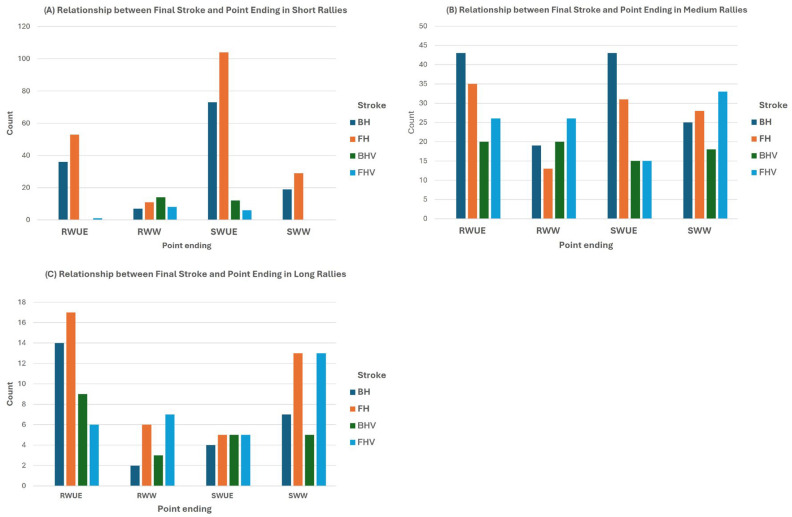
Relationship between point ending and final stroke across different rally types. (**A**) Relationship between Final Stroke and Point Ending in Short Rallies, (**B**) Relationship between Final Stroke and Point Ending in Medium Rallies, (**C**) Relationship between Final Stroke and Point Ending in Long Rallies.

**Table 1 jfmk-10-00020-t001:** Description of OI-PICKLEBALL-S23.

Criteria	Code	Description
Service	FS	First service
	SF	Service fault
Rally length (the serve shot is counted)	SH	Short rally (1–4 shots).
	MD	Medium rally (5–8 shots).
	LN	Long rally (9+ shots).
Strike zone (see Figure 1)	SZ1	Non-volley zone
	SZ2	Mid-court zone
	SZ3	Back court zone, including the baseline
	SZ4	Deep court zone, behind the baseline
	SZ	Service zone
Finish zone (see Figure 1)	FZ1	Left front zone
	FZ2	Right front zone
	FZ3	Left mid-court zone
	FZ4	Right mid-court zone
	FZ5	Left back court zone
	FZ6	Right back court zone
	NT	Net shot
	OUT	Shot out
Winner	SW	The point is won by the server
	RW	The point is won by the returner
Point ending	SWW	Server wins with a winner or a forced error
	SWUE	Server wins with an unforced error by the opponent
	RWW	Receiver wins with a winner or a forced error
	RWUE	Receiver wins with an unforced error by the opponent
Final stroke	ACE	Direct serve
	FH	Forehand
	BH	Backhand
	FHV	Forehand volley
	BHV	Backhand volley
	SM	Smash
	LB	Lob
	DS	Drop shot
	SC	Change of service due to an error in the service
	OT	Other type of stroke

**Table 2 jfmk-10-00020-t002:** Descriptive analysis of the study.

Criteria	Code	*n*	%	*χ*^2^ Goodness-of-Fit	Criteria	Code	*n*	%	*χ*^2^ Goodness-of-Fit
Service	FS	934	97.7	*χ*^2^ = 870.025	Winner	RW	429	44.9	*χ*^2^ = 10.046
	SF	22	2.3	*p* < 0.000		SW	527	55.1	*p* < 0.002
Rally length	LN	129	13.5	*χ*^2^ = 170.161	Point ending	RWUE	285	29.8	*χ*^2^ = 82.268
	MD	425	44.5	*p* < 0.000		RWW	144	15.1	*p* < 0.000
	SH	402	42.1			SWUE	324	33.9	
Strike zone	SZ1	6	0.6	*χ*^2^ = 612.839		SWW	203	21.2	
	SZ2	342	35.8	*p* < 0.000	Final stroke	ACE	0	0	*χ*^2^ = 1314.504
	SZ3	215	22.5			BH	292	30.5	*p* < 0.000
	SZ4	369	38.6			BHV	121	12.7	
	SZ	24	2.5			DS	1	0.1	
The finish zone	NT	259	27.1	*χ*^2^ = 856.686		FH	345	36.1	
	OUT	350	36.6	*p* < 0.000		FHV	146	15.3	
	FZ1	17	1.8			LB	5	0.5	
	FZ2	8	0.8			OT	4	0.4	
	FZ3	73	7.6			SC	23	2.4	
	FZ4	84	8.8			SM	19	2.0	
	FZ5	64	6.7						
	FZ6	101	10.6						

**Table 3 jfmk-10-00020-t003:** T-Patterns with short rallies depending on the point winner (*n* = 380).

Game Pattern FH	*n*	%	Game Pattern BH	*n*	%	Game Pattern FHV	*n*	%	Game Pattern BHV	*n*	%
SW-SH-FH	133	35	SW-SH-BH	92	24.2	SW-SH-FHV	6	1.6	SW-SH-BHV	12	3.2
SW-SH-SZ2-FH	0	0	SW-SH-SZ2-BH	5	1.3	SW-SH-SZ2-FHV	6	1.6	SW-SH-SZ2-BHV	12	3.2
SW-SH-SZ2-SWW-FH	0	0	SW-SH-SZ2-SWW-BH	0	0	SW-SH-SZ2-SWW-FHV	0	0	SW-SH-SZ2-SWW-BHV	0	0
SW-SH-SZ2-SWUE-FH	0	0	SW-SH-SZ2-SWUE-BH	5	1.3	SW-SH-SZ2-SWUE-FHV	6	1.6	SW-SH-SZ2-SWUE-BHV	12	3.2
SW-SH-SZ3-FH	16	4.2	SW-SH-SZ3-BH	15	3.9	SW-SH-SZ3-FHV	0	0	SW-SH-SZ3-BHV	0	0
SW-SH-SZ3-SWW-FH	10	2.6	SW-SH-SZ3-SWW-BH	8	2.1	SW-SH-SZ3-SWW-FHV	0	0	SW-SH-SZ3-SWW-BHV	0	0
SW-SH-SZ3-SWUE-FH	6	1.6	SW-SH-SZ3-SWUE-BH	7	1.8	SW-SH-SZ3-SWUE-FHV	0	0	SW-SH-SZ3-SWUE-BHV	0	0
SW-SH-SZ4-FH	117	30.8	SW-SH-SZ4-BH	72	18.9	SW-SH-SZ4-FHV	0	0	SW-SH-SZ4-BHV	0	0
SW-SH-SZ4-SWW-FH	19	5	SW-SH-SZ4-SWW-BH	11	2.9	SW-SH-SZ4-SWW-FHV	0	0	SW-SH-SZ4-SWW-BHV	0	0
SW-SH-SZ4-SWUE-FH	98	25.8	SW-SH-SZ4-SWUE-BH	61	16.1	SW-SH-SZ4-SWUE-FHV	0	0	SW-SH-SZ4-SWUE-BHV	0	0
RW-SH-FH	64	16.8	RW-SH-BH	43	11.3	RW-SH-FHV	9	2.4	RW-SH-BHV	14	3.7
RW-SH-SZ2-FH	0	0	RW-SH-SZ2-BH	0	0	RW-SH-SZ2-FHV	8	2.1	RW-SH-SZ2-BHV	14	3.7
RW-SH-SZ2-RWW-FH	0	0	RW-SH-SZ2-RWW-BH	0	0	RW-SH-SZ2-RWW-FHV	8	2.1	RW-SH-SZ2-RWW-BHV	14	3.7
RW-SH-SZ2-RWUE-FH	0	0	RW-SH-SZ2-RWUE-BH	0	0	RW-SH-SZ2-RWUE-FHV	0	0	RW-SH-SZ2-RWUE-BHV	0	0
RW-SH-SZ3-FH	23	6.1	RW-SH-SZ3-BH	12	3.2	RW-SH-SZ3-FHV	1	0.3	RW-SH-SZ3-BHV	0	0
RW-SH-SZ3-RWW-FH	5	1.3	RW-SH-SZ3-RWW-BH	4	1.1	RW-SH-SZ3-RWW-FHV	1	0.3	RW-SH-SZ3-RWW-BHV	0	0
RW-SH-SZ3-RWUE-FH	18	4.7	RW-SH-SZ3-RWUE-BH	8	2.1	RW-SH-SZ3-RWUE-FHV	0	0	RW-SH-SZ3-RWUE-BHV	0	0
RW-SH-SZ4-FH	40	10.5	RW-SH-SZ4-BH	31	8.2	RW-SH-SZ4-FHV	0	0	RW-SH-SZ4-BHV	0	0
RW-SH-SZ4-RWW-FH	6	1.6	RW-SH-SZ4-RWW-BH	3	0.8	RW-SH-SZ4-RWW-FHV	0	0	RW-SH-SZ4-RWW-BHV	0	0
RW-SH-SZ4-RWUE-FH	34	8.9	RW-SH-SZ4-RWUE-BH	28	7.4	RW-SH-SZ4-RWUE-FHV	0	0	RW-SH-SZ4-RWUE-BHV	0	0

Note: SW: the server wins; RW: the receiver wins; SH: short rally; SZ2: mid-court striking zone; SZ3: back court striking zone, including the baseline; SZ4: deep court striking zone, behind the baseline; SWW: the server wins with a winner or a forced error; SWUE: the server wins with an unforced error by the opponent; RWW: the receiver wins with a winner or a forced error; RWUE: the receiver wins with an unforced error by the opponent; FH: forehand; BH: backhand; FHV: forehand volley; BHV: backhand volley.

**Table 4 jfmk-10-00020-t004:** T-Patterns with medium rallies, depending on the point winner (*n* = 425).

Game Pattern FH	*n*	%	Game Pattern BH	*n*	%	Game Pattern FHV	*n*	%	Game Pattern BHV	*n*	%
SW-MD-FH	59	13.9	SW-MD-BH	68	16	SW-MD-FHV	48	11.3	SW-MD-BHV	33	7.8
SW-MD-SZ2-FH	11	2.6	SW-MD-SZ2-BH	12	2.8	SW-MD-SZ2-FHV	43	10.1	SW-MD-SZ2-BHV	32	7.5
SW-MD-SZ2-SWW-FH	6	1.4	SW-MD-SZ2-SWW-BH	8	1.9	SW-MD-SZ2-SWW-FHV	30	7.1	SW-MD-SZ2-SWW-BHV	18	4.2
SW-MD-SZ2-SWUE-FH	5	1.2	SW-MD-SZ2-SWUE-BH	4	0.9	SW-MD-SZ2-SWUE-FHV	13	3.1	SW-MD-SZ2-SWUE-BHV	14	3.3
SW-MD-SZ3-FH	30	7.1	SW-MD-SZ3-BH	30	7.1	SW-MD-SZ3-FHV	5	1.2	SW-MD-SZ3-BHV	0	0
SW-MD-SZ3-SWW-FH	18	4.2	SW-MD-SZ3-SWW-BH	14	3.3	SW-MD-SZ3-SWW-FHV	3	0.7	SW-MD-SZ3-SWW-BHV	0	0
SW-MD-SZ3-SWUE-FH	12	2.8	SW-MD-SZ3-SWUE-BH	16	3.8	SW-MD-SZ3-SWUE-FHV	2	0.5	SW-MD-SZ3-SWUE-BHV	0	0
SW-MD-SZ4-FH	18	4.2	SW-MD-SZ4-BH	25	5.9	SW-MD-SZ4-FHV	0	0	SW-MD-SZ4-BHV	1	0.2
SW-MD-SZ4-SWW-FH	4	0.9	SW-MD-SZ4-SWW-BH	3	0.7	SW-MD-SZ4-SWW-FHV	0	0	SW-MD-SZ4-SWW-BHV	0	0
SW-MD-SZ4-SWUE-FH	14	3.3	SW-MD-SZ4-SWUE-BH	22	5.2	SW-MD-SZ4-SWUE-FHV	0	0	SW-MD-SZ4-SWUE-BHV	1	0.2
RW-MD-FH	48	11.3	RW-MD-BH	62	14.6	RW-MD-FHV	52	12.2	RW-MD-BHV	40	9.4
RW-MD-SZ2-FH	8	1.9	RW-MD-SZ2-BH	14	3.3	RW-MD-SZ2-FHV	47	11.1	RW-MD-SZ2-BHV	36	8.5
RW-MD-SZ2-RWW-FH	3	0.7	RW-MD-SZ2-RWW-BH	8	1.9	RW-MD-SZ2-RWW-FHV	25	5.9	RW-MD-SZ2-RWW-BHV	18	4.2
RW-MD-SZ2-RWUE-FH	5	1.2	RW-MD-SZ2-RWUE-BH	6	1.4	RW-MD-SZ2-RWUE-FHV	22	5.2	RW-MD-SZ2-RWUE-BHV	18	4.2
RW-MD-SZ3-FH	15	3.5	RW-MD-SZ3-BH	26	6.1	RW-MD-SZ3-FHV	5	1.2	RW-MD-SZ3-BHV	4	0.9
RW-MD-SZ3-RWW-FH	5	1.2	RW-MD-SZ3-RWW-BH	10	2.4	RW-MD-SZ3-RWW-FHV	1	0.2	RW-MD-SZ3-RWW-BHV	0	0
RW-MD-SZ3-RWUE-FH	10	2.4	RW-MD-SZ3-RWUE-BH	16	3.8	RW-MD-SZ3-RWUE-FHV	4	0.9	RW-MD-SZ3-RWUE-BHV	0	0
RW-MD-SZ4-FH	25	5.9	RW-MD-SZ4-BH	22	5.2	RW-MD-SZ4-FHV	0	0	RW-MD-SZ4-BHV	0	0
RW-MD-SZ4-RWW-FH	5	1.2	RW-MD-SZ4-RWW-BH	1	0.2	RW-MD-SZ4-RWW-FHV	0	0	RW-MD-SZ4-RWW-BHV	0	0
RW-MD-SZ4-RWUE-FH	20	4.7	RW-MD-SZ4-RWUE-BH	21	4.9	RW-MD-SZ4-RWUE-FHV	0	0	RW-MD-SZ4-RWUE-BHV	0	0

Note: SW: the server wins; RW: the receiver wins; MD: medium rally; SZ2: mid-court striking zone; SZ3: back court striking zone, including the baseline; SZ4: deep court striking zone, behind the baseline; SWW: the server wins with a winner or a forced error; SWUE: the server wins with an unforced error by the opponent; RWW: the receiver wins with a winner or a forced error; RWUE: the receiver wins with an unforced error by the opponent; FH: forehand; BH: backhand; FHV: forehand volley; BHV: backhand volley.

## Data Availability

The data presented in this study are openly available in FigShare at doi https://doi.org/10.6084/m9.figshare.28124738.

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
