# Peer review of "Technical and Tactical Performance in Women’s Singles Pickleball: A Notational Analysis of Key Match Indicators"

_jfmk, 2025, doi:10.3390/jfmk10010020_

Round 1

Reviewer 1 Report

Comments and Suggestions for Authors

The studies need some adjustments, not many, but necessary to make the research clearer in its explanation.

Reviewer 2 Report

Comments and Suggestions for Authors

Line 29

Pickleball, although a relatively young sport

Comment 1

Can you be more specific about the age of the sport?

Comment 2

Little is known about this sport, so more information should be given in the introduction. What place does it occupy among racquet sports? In the articles on pickleball, I have seen that there is a lot of talk about older adults?

Line 91

The participants included professional players who advanced to at least the semifinal stage in these events.

Comment 3

Are they really professionals (they make a living from sports) or elite players? Is it strange that a new sport has a professional league?

Line 97-101

The observation tool utilized in this study was the OI-PICKLEBALL-S23, a category system specifically designed for analyzing pickleball during the 2023 season [23]. This instrument had been previously applied in a study focusing on the men's category and underwent a validation process to ensure its reliability and alignment with the research objectives.

Comment 4

You didn't specify in which study it was validated?

Line 323-325

To optimize performance in pickleball, training for both genders should focus on improving these key shots and developing strategies that maximize their effectiveness during play.

Comment 5

This should be written in more detail, a little more about key points and strategies?

Round 2

Reviewer 1 Report

Comments and Suggestions for Authors

The authors made the necessary adjustments.